# Enhanced Parameter Estimation with Periodically Driven Quantum Probe

**DOI:** 10.3390/e23101333

**Published:** 2021-10-12

**Authors:** Peter A. Ivanov

**Affiliations:** Department of Physics, St. Kliment Ohridski University of Sofia, James Bourchier 5 Blvd, 1164 Sofia, Bulgaria; pivanov@phys.uni-sofia.bg

**Keywords:** quantum sensing, trapped ions, periodic modulation

## Abstract

I propose a quantum metrology protocol for measuring frequencies and weak forces based on a periodic modulating quantum Jahn–Teller system composed of a single spin and two bosonic modes. I show that, in the first order of the frequency drive, the time-independent effective Hamiltonian describes spin-dependent interaction between the two bosonic modes. In the limit of high-frequency drive and low bosonic frequency, the quantum Jahn–Teller system exhibits critical behavior which can be used for high-precision quantum estimation. A major advantage of the scheme is the robustness of the system against spin decoherence, which allows it to perform parameter estimation with measurement time not limited by spin dephasing.

## 1. Introduction

Over the last few years, there has been considerable interest in the development of high-precision quantum metrology with strongly correlated quantum systems [1,2]. One way to improve the precision of parameter estimation is to use entangled states [3,4,5]. Indeed, entangled states may yield a favorable scaling in the parameter precision measurement compared to what is possible with uncorrelated states. Another approach for high-precision quantum metrology is based on a probe system which exhibits a quantum phase transition [6,7,8,9]. Such a criticality-enhanced quantum metrology can be used to perform a high-precision measurement of the control parameter close to the quantum phase transition. Recently, an experimental realization of quantum sensor with sensitivity enhanced by quantum criticality was demonstrated with a Bose-Einstein condensate [10]. Usually, the existence of the quantum phase transition requires a thermodynamic limit, where the number of constituents goes to infinity. A different class of phase transitions was introduced in an interacting system of single-mode cavity field and two-level atom, where the thermodynamic limit requires the cavity frequency in units of atomic transition frequency to tend to zero [11,12]. An enhanced parameter estimation was proposed with such finite size critical quantum optical system for high-precision force measurements [13,14,15] or frequency measurements [8,9]. The corresponding quantum Fisher information diverges by approaching the critical coupling, indicating that the finite size quantum optical system becomes sensitive to infinitely small variation of the parameter of interest.

In this work, I consider the quantum metrology application of the finite size periodic modulating quantum system consisting of interacting single spin and two bosonic modes described by the quantum Jahn–Teller (JT) model. In my scheme, spin-boson couplings are periodically modulated, which drives the system into the regime described by the effective Hamiltonian. I show that, under the high-frequency drive, the spin evolution is suppressed; thereby, it can be adiabatically eliminated from the dynamics. In the first order of the frequency drive, the effective Hamiltonian describes spin-dependent interaction between two bosonic modes. I show that, in the limit of high-frequency drive and low bosonic frequency, the effective model exhibits critical behavior which can be used for high-precision quantum metrology.

Furthermore, I include the dissipative processes which affect the two bosonic modes. In that case, the balance of the periodic drive and the losses of bosonic excitations drives the system into the nonequilibrium steady state. I show that the time-periodic driven dissipative dynamics is described in terms of an effective Liouvillian. I characterize the steady state density operator in terms of its first and second moments. In the high-frequency drive regime, the density matrix reviews a non-analytical behavior. I derive expression for the quantum Fisher information and show that it diverges close to the critical point. I also consider the decoherence process of loss of spin coherence caused, for example, by fluctuating magnetic fields. Such a spin dephasing is the major source of loss of contrast which reduces the optimal precision of frequency measurements [16]. Remarkably, under the condition of high-frequency drive the resulting effective Liouvillian is diagonal in the spin basis. Consequently, the time evolution of our periodic modulating JT system is immune against spin decoherence. This allows for performing frequency estimation with measurement time, which is not limited by spin dephasing.

Finally, I provide a scheme for the physical implementation of our periodically driven dissipative JT model with trapped ions. In the proposed realization, the two local phonons along the spatial *x*-*y* directions correspond to the the bosonic modes. Bichromatic laser fields with time-dependent periodic intensity are used to couple the internal ion’s spin states and the two phonons, which provide the desired JT spin-boson coupling. I show that the sympathetic cooling of an auxiliary ion can be used to create the dissipative dynamics of the two bosonic modes.

The paper is organized as follows: In Section 2, the periodic modulating dissipative JT model is introduced. The dynamics of the JT model in terms of an effective Liouvillian is discussed. It is shown that, in the limit of high-frequency drive, the spin dynamics are suppressed, and the effective Hamiltonian describes two interacting bosonic modes. In Section 3, I consider the coherent evolution of the periodic modulating JT system. I show that, for high-frequency drive and low bosonic frequency, the signal-to-noise ratio is improved, which allows for performing a high-precision frequency estimation. In Section 4, I discuss the steady-state density matrix of the periodic modulating dissipative JT model. The physical realization of the model is provided in Section 5. Finally, the conclusions are presented in Section 6.

## 2. Model

### 2.1. Periodic Modulating Dissipative Jahn–Teller Interaction

I consider in the following a quantum system of two bosonic modes which interact with a single spin via periodic modulating dipolar coupling. Let

(1)
$${\hat{H}}_{0}=\omega_{x}{\hat{a}}_{x}^{†}{\hat{a}}_{x}+\omega_{y}{\hat{a}}_{y}^{†}{\hat{a}}_{y}+\frac{\Delta }{2}\sigma_{z}$$

denote the time-independent Hamiltonian, which describes the quantum system in the absence of periodic driving. Here, 
${\hat{a}}_{\beta }$
 and 
${\hat{a}}_{\beta }^{†}$
 (
β=x,y
) are the annihilation and creation operators of bosonic excitation with frequency 
$\omega_{\beta }$
 in mode 
β
. The single spin is described with the Pauli matrices 
$\sigma_{x,y,z}$
, and 
Δ
 stands for the transition spin frequency. The effect of the driving is represented by a time-dependent part of the total Hamiltonian

(2)
$$\hat{H}\left(t\right)={\hat{H}}_{0}+{\hat{H}}_{d}\left(t\right),$$

where we assume

(3)
$${\hat{H}}_{d}\left(t\right)=2g_{x}\cos\left(\Phi t\right)\sigma_{x}\left({\hat{a}}_{x}^{†}+{\hat{a}}_{x}\right)+2g_{y}\sin\left(\Phi t\right)\sigma_{y}\left({\hat{a}}_{y}^{†}+{\hat{a}}_{y}\right),$$

with 
$g_{\beta }$
 being the spin-boson coupling, and 
Φ
 the driving frequency. In the absence of driving, the Hamiltonian (Equation 3) describes dipolar JT interaction between a single spin with two vibrational modes. Such a JT coupling was first introduce to explain distortions and nondegenerate energy levels in molecules and condensed quantum systems [17]. Various quantum optical systems have been proposed to simulate the JT model, including, for example, cavity QED systems [18] and spin-orbit-coupled Bose-Einstein condensates [19]. Recently, the Rydberg trapped ion realization of the Jahn–Teller modes was proposed in Reference [20].

The Fourier series of (Equation 3) can be written as 
${\hat{H}}_{d}\left(t\right)=e^{i\Phi t}\hat{v}+e^{-i\Phi t}v^{†}$
, where 
$\hat{v}=g_{x}\sigma_{x}\left({\hat{a}}_{x}^{†}+{\hat{a}}_{x}\right)-ig_{y}\sigma_{y}\left({\hat{a}}_{y}^{†}+{\hat{a}}_{y}\right)$
, which ensures that 
${\hat{H}}_{d}\left(t+T\right)={\hat{H}}_{d}\left(t\right)$
; hence, 
$\hat{H}\left(t+T\right)=\hat{H}\left(t\right)$
, with *T* being the driving period.

To study driven-dissipative system, we consider the density operator 
$\hat{\rho }\left(t\right)$
 whose dynamics is governed by the Lindblad equation [21]

(4)
$$\partial_{t}\hat{\rho }\left(t\right)=\hat{L}\left(t\right)\hat{\rho }\left(t\right)=-i\left[\hat{H}\left(t\right),\hat{\rho }\left(t\right)\right]+\sum_{j}\hat{D}\left[{\hat{L}}_{j}\right]\hat{\rho }\left(t\right).$$


Here, 
$\hat{L}\left(t\right)$
 is a time-dependent Liouvillian superoperator, while the term 
$\hat{D}\left[{\hat{L}}_{j}\right]$
 is the Lindblad dissipator, whose action is given by

(5)
$$\hat{D}\left[{\hat{L}}_{j}\right]\hat{\rho }\left(t\right)=2{\hat{L}}_{j}\hat{\rho }\left(t\right){\hat{L}}_{j}^{†}-{\hat{L}}_{j}^{†}{\hat{L}}_{j}\hat{\rho }\left(t\right)-\hat{\rho }\left(t\right){\hat{L}}_{j}^{†}{\hat{L}}_{j},$$

where 
${\hat{L}}_{j}$
 are the jump operators which describe how the environment affect the system evolution. In this work, I consider the process of loss of bosonic excitations where the quantum jump operators are given by 
${\hat{L}}_{1}=\sqrt{\gamma_{x}}{\hat{a}}_{x}$
, and, respectively, 
${\hat{L}}_{2}=\sqrt{\gamma_{y}}{\hat{a}}_{y}$
, with 
$\gamma_{x,y}$
 being the respective bosonic decay rates. I also discuss the effect of spin dephasing on the spin-dependent bosonic modes evolution. As we will see below, due to the condition of high frequency drive, the effective time-averaged dynamics is diagonal in the spin basis. As a result of that, the quantum JT system becomes immune against the spin dephasing, which can have significant impact on the high-precision quantum estimation.

### 2.2. Time-Average Dynamics

In the following, I explore the nonequilibrium steady state which emerges in a balance between the periodic drive and boson dissipation. The physical properties of the periodically driven quantum system can be described in terms of effective Hamiltonian, which reflects the periodic driving according to the Floquet theorem. For closed driven quantum systems which are not subject to dissipative processes, the time-evolution can be split into the product of kick operators which describes the residual micromotion and time-independent evolution dictated by the effective Hamiltonian [22]. Recently, an expression for the nonequilibrium steady state in the limit of the high-frequency expansion of the Lindblad equation was derived [23]. Assuming that the system is prepared initially in a state with 
$\hat{\rho }\left(0\right)$
 the density operator at time *t* can be written as 
$\hat{\rho }\left(t\right)=e^{\hat{G}\left(t\right)}e^{t{\hat{L}}_{\mathrm{eff}}}e^{-\hat{G}\left(0\right)}\hat{\rho }\left(0\right)$
. In the leading order of 
$\Phi^{-1}$
, the time-independent effective Liouvillian is given by [23]

(6)
$$\begin{array}{ccc} & & {\hat{L}}_{\mathrm{eff}}\hat{\rho }=-i\left[{\hat{H}}_{\mathrm{eff}},\hat{\rho }\right]+\sum_{\beta =x,y}\gamma_{\beta }\hat{D}\left[{\hat{a}}_{\beta }\right]\hat{\rho }\left(t\right),\\ & & {\hat{H}}_{\mathrm{eff}}={\hat{H}}_{0}+\frac{1}{\Phi }\left[\hat{v},{\hat{v}}^{†}\right]+O\left(\Phi^{-2}\right), \end{array}$$

where 
${\hat{H}}_{\mathrm{eff}}$
 is the time-independent effective Hamiltonian. Finally, the period time-dependent micromotion operator is given by 
$\hat{G}\left(t\right)\hat{\rho }=\Phi^{-1}\{\left[\hat{v},\hat{\rho }\right]e^{i\Phi t}+\left[{\hat{v}}^{†},\hat{\rho }\right]e^{-\Phi t}$
}.

Using (Equation 2), we find that the effective Hamiltonian becomes

(7)
$$\begin{array}{ccc} {\hat{H}}_{\mathrm{eff}} & = & \omega_{x}{\hat{a}}_{x}^{†}{\hat{a}}_{x}+\omega_{y}{\hat{a}}_{y}^{†}{\hat{a}}_{y}+\frac{\Delta }{2}\sigma_{z}-\frac{4g_{x}g_{y}}{\Phi }\sigma_{z}\left({\hat{a}}_{x}^{†}+{\hat{a}}_{x}\right)\left({\hat{a}}_{y}^{†}+{\hat{a}}_{y}\right)\\ & & +O\left(\Phi^{-2}\right), \end{array}$$

where we assume 
$\Phi \gg g_{\beta },\omega_{\beta },\Delta ,\gamma_{\beta }$
 (high-frequency drive regime). We observe that the effective Hamiltonian is diagonal in the spin basis. Moreover, the periodic driving causes spin-dependent coupling between the two *x* and *y* bosonic modes, which is of order of 
$\Phi^{-1}$
; thus, it cannot be neglected.

In the following, I provide the diagonalization of the effective Hamiltonian (Equation 7). I show that, in the high-frequency drive regime and low bosonic frequencies, the effective model exhibits critical behavior which can be used for high-precision quantum metrology.

## 3. Quantum Metrology with Periodic Modulating Quantum System: Coherent Evolution

Before discussing the dissipative dynamics, I consider first the eigenfrequencies of the Hamiltonian. Because (Equation 7) is quadratic in the bosonic operators, it can be exactly diagonalized. Hereafter, I assume that the spin is initially prepared in the state 
$|\psi_{s}\rangle =|↑\rangle$
, where 
$\sigma_{z}|↑\rangle =|↑\rangle$
. Then, performing generalized Bogoliubov transformation (see the Appendix A for an overview of the derivation), the effective Hamiltonian is transformed in a canonical form, 
${\hat{H}}_{\mathrm{eff}}=\omega \sum_{\alpha =1}^{2}\nu_{\alpha }{\hat{d}}_{\alpha }^{†}{\hat{d}}_{\alpha }$
, where 
$\nu_{1}=\sqrt{1-\lambda^{2}}$
 and 
$\nu_{2}=\sqrt{1+\lambda^{2}}$
 are the eigenfrequencies (we set 
$\omega_{x,y}=\omega$
), with 
$\lambda =\sqrt{\frac{8g_{x}g_{y}}{\omega \Phi }}$
 being the dimensionless coupling parameter. The energy gap tends to zero when 
λ→1
, which signals the existence of critical point and emergence of quantum phase transition [24]. Such finite size quantum phase transition was discussed in the context of the quantum Rabi model [11], where the dimensionless parameter 
$\eta_{\mathrm{qr}}=\omega /\Omega$
 is introduced. In the limit 
$\eta_{\mathrm{qr}}\to 0$
, the quantum Rabi model exhibits a phase transition, which was recently experimentally observed [12]. Here, one can define the ratio 
$\eta_{\mathrm{pm}}=\omega /\Phi$
 such that, in the limit 
$\eta_{\mathrm{pm}}\to 0$
, the periodically driven quantum JT system exhibits critical behavior at 
$\lambda_{c}=1$
. In Figure 1, the exact time-evolution is shown of the position quadrature 
$\langle {\hat{q}}_{1}\rangle$
 and its variance 
$\Delta {\hat{q}}_{1}=\sqrt{\langle {\hat{q}}_{1}^{2}\rangle -{\langle {\hat{q}}_{1}\rangle }^{2}}$
 using the time-dependent Hamiltonian (Equation 2). Here, 
$\hat{q}=\left\{\hat{x},{\hat{p}}_{x},\hat{y},{\hat{p}}_{y}\right\}$
 is the bosonic quadrature operator, where 
$\hat{x}=\left({\hat{a}}_{x}^{†}+{\hat{a}}_{x}\right)$
, 
${\hat{p}}_{x}=i\left({\hat{a}}_{x}^{†}-{\hat{a}}_{x}\right)$
 and, respectively, 
$\hat{y}=\left({\hat{a}}_{y}^{†}+{\hat{a}}_{y}\right)$
, 
${\hat{p}}_{y}=i\left({\hat{a}}_{y}^{†}-{\hat{a}}_{y}\right)$
 are the position and momentum quadrature operators for the two bosonic modes. I compare the numerical result with the analytical expressions

(8)
$$\begin{array}{ccc} & & \langle \hat{x}\left(t\right)\rangle =\frac{\sin(\omega \nu_{1}t)}{\nu_{1}},\\ & & \Delta \hat{x}{\left(t\right)}^{2}=\frac{1}{4\nu_{1}^{2}\nu_{2}^{2}}\left\{6-{\left(\nu_{1}^{2}-\nu_{2}^{2}\right)}^{2}+\left(\nu_{1}^{2}-2{\left(1-\nu_{1}^{2}\right)}^{2}\right)\right)\\ & & \;\;\;\times \cos\left(2\omega \nu_{1}t\right)+\left(\nu_{2}^{2}-2{\left(1-\nu_{2}^{2}\right)}^{2}\right))\cos\left(2\omega \nu_{2}t\right)\}, \end{array}$$

which are derived from the Heisenberg equation of motion for initial two bosonic modes state 
$|\psi (0)\rangle =|\psi_{x}\rangle ⊗|\psi_{y}\rangle$
, where 
$|\psi_{\beta }\rangle =2^{-1/2}\left(|0_{\beta }\rangle +i|1_{\beta }\rangle \right)$
 (see the Appendix A for more details). Here, 
$|n_{\beta }\rangle$
 is the Fock state of the bosonic mode with occupation number 
$n_{\beta }$
. As we see, very good agreement between the exact and the analytical results is observed, which indicates that the time-evolution is mainly dictated by the effective Hamiltonian (Equation 7).

Measuring the position quadrature 
$\langle \hat{x}\left(t\right)\rangle$
, one can estimate, for example, the bosonic frequency 
ω
. In order to quantify the sensitivity in the frequency estimation, we use fidelity susceptibility 
$F_{x}\left(\omega \right)=\frac{\partial_{\omega }\langle \hat{x}\left(t\right)\rangle }{\Delta \hat{x}\left(t\right)}$
[1]. The shot-noise limited sensitivity in the estimation of 
ω
 from the measured signal 
$\langle \hat{x}\left(t\right)\rangle$
 is 
$\delta \omega =1/F_{x}\left(\omega \right)$
. In Figure 2, we plot the fidelity susceptibility as a function of time for different couplings 
λ
. As 
λ
 increases toward the critical coupling 
$\lambda_{c}=1$
, the sensitivity in the frequency estimation is improved. Indeed, using (Equation 8), it is straightforward to show that, at time 
$t^{*}=\pi /\omega \nu_{1}$
 and 
λ
 approaching 
$\lambda_{c}$
, we have 
$\partial_{\omega }\langle \hat{x}\left(t^{*}\right)\rangle ∼\left(\pi /\sqrt{32}\omega \right){\left(1-\lambda \right)}^{-3/2}$
 and, respectively, 
$\Delta \hat{x}{\left(t^{*}\right)}^{2}∼\left(13+3\cos\left(2\pi \nu_{2}/\nu_{1}\right)\right)/8$
. Therefore, minimizing the position variance, the uncertainty in the boson frequency estimation scales as 
$\delta \omega ∼\frac{2\sqrt{10}\omega }{\pi }{\left(1-\lambda \right)}^{3/2}$
, which implies that arbitrarily large frequency estimation precision can be achieved close to the criticality.

Our technique can also be applied to the measurement of the spin frequency 
Δ
. The effect of 
Δ
 on the spin-boson interaction is of order of 
$\Phi^{-2}$
 (see the Appendix A). Including such terms, I find that the eigenfrequencies of the effective Hamiltonian are modified according to 
$\nu_{1}\left(\epsilon \right)=\sqrt{1-\lambda_{+}^{2}\left(\epsilon \right)}$
 and 
$\nu_{2}\left(\epsilon \right)=\sqrt{1+\lambda_{-}^{2}\left(\epsilon \right)}$
, where the couplings are 
$\lambda_{\pm }\left(\epsilon \right)=\sqrt{\frac{8g^{2}}{\omega \Phi }\left(1\pm \epsilon \right)}$
 (we assume 
$g=g_{\beta }$
), and 
ϵ=Δ/Φ≪1
. Then, using (Equation 8), I obtain 
$\partial_{\epsilon }\langle \hat{x}\left(t^{*}\right)\rangle ∼\left(4g^{2}\pi /\omega \Phi \right){\left(1-\lambda_{+}^{2}\left(\epsilon \right)\right)}^{-3/2}$
 and 
$\Delta \hat{x}{\left(t^{*}\right)}^{2}∼\left(13+\epsilon +\left(3-\epsilon \right)\cos\left(2\pi \nu_{2}\left(\epsilon \right)/\nu_{1}\left(\epsilon \right)\right)\right)/8$
, where 
$t^{*}=\pi /\omega \nu_{1}\left(\epsilon \right)$
. For the spin frequency uncertainty estimation, I obtain 
$\delta \epsilon =1/F_{x}\left(\epsilon \right)∼\left(\sqrt{5}/\pi \right){\left(1-\lambda_{+}\left(\epsilon \right)\right)}^{3/2}$
, which again becomes infinitesimally small close to the critical point.

In the following, I discuss the the effect of dissipation of the bosonic excitations. In that case, the time-periodic drives, and the loss of bosonic excitations leads to nonequilibrium steady state, which we describe in terms of an effective time-independent Liouvillian.

## 4. Quantum Metrology with Periodic Modulating Dissipative Quantum System

Let us now consider the potential quantum metrology application of our periodically driven dissipative JT quantum system. The interplay between the dissipative dynamics and the coherent driving leads to emergence of nonequilibrium steady state. Such a steady state density matrix may exhibit non-analytical behavior at the criticality [25]. Indeed, the critical dissipative phase transitions are characterized by a nonanalytical change of the steady state and can be used to enhance the sensitivity of single and multi parameter estimation close to a quantum critical point [15,26,27].

Consider the strong driving regime, where the effect of the time-dependent micro-motion term can be neglected; thereby, the time-evolution of the system is mainly dictated by the effective time-independent Liouvillian 
${\hat{L}}_{\mathrm{eff}}\hat{\rho }$
. Hereafter, I also assume that a force displacement term 
${\hat{H}}_{f}=\left(f/2\right)\left({\hat{a}}_{x}^{†}+{\hat{a}}_{x}\right)$
 with magnitude *f* is applied along the *x* direction which displaces the respective bosonic mode. Since the displacement term is time-independent, it does not affect the time-average dynamics so that the total effective Hamiltonian becomes 
${\hat{H}}_{T}={\hat{H}}_{\mathrm{eff}}+{\hat{H}}_{f}$
.

As time increases, the system approaches the steady state with density matrix 
${\hat{\rho }}_{\mathrm{ss}}$
. Because the coherent, as well as the dissipative, dynamics are quadratic in the bosonic operators the steady state is in a Gaussian form, so that it can be completely characterized with the first and the second moments [28]. Let us define the symmetric covariance matrix, whose elements are

(9)
$$V{\left({\hat{\rho }}_{\mathrm{ss}}\right)}_{kl}=\frac{1}{2}\langle {\hat{q}}_{k}{\hat{q}}_{l}+{\hat{q}}_{l}{\hat{q}}_{k}\rangle -d_{k}d_{l},$$

where 
$d=\langle \hat{q}\rangle$
 is the mean displacement vector, and all expectation values are taken with respect to the steady state 
${\hat{\rho }}_{\mathrm{ss}}$
.

In Figure 3a, the numerical result is shown for the covariance elements 
$V_{11}$
 and 
V22
 as a function of the coupling strength 
$g=g_{\beta }$
. In the steady state regime, the matrix elements are given by

(10)
$$V_{11}=\frac{2\lambda_{c}^{4}-\lambda^{4}}{2(\lambda_{c}^{4}-\lambda^{4})},\;V_{22}=\frac{2\lambda_{c}^{4}+\left(\lambda_{c}^{2}-3\right)\lambda^{4}}{2(\lambda_{c}^{4}-\lambda^{4})},$$

where I have assumed 
$\omega_{\beta }=\omega$
. Due to the symmetry, we have 
$V_{11}=V_{33}$
 and 
$V_{22}=V_{44}$
. All other covariance matrix elements are presented in the Appendix A. One can identify two couplings defined by 
$\lambda_{\pm c}^{2}=\left(1+\gamma^{2}/\omega^{2}\right){\left(1\pm \epsilon \right)}^{-1}$
. Up to first order of 
$\Phi^{-1}$
, we have 
$\lambda_{\pm c}^{2}=\lambda_{c}^{2}$
, with 
$\lambda_{c}^{2}=1+\gamma^{2}/\omega^{2}$
 being the critical coupling. We see that, for strong periodic drive, the effect of the micromotion term can be neglected such that the behavior of the system is dictated by the effective Liouvillian (Equation 6). Increasing the spin-boson coupling increases the covariance elements, as well, and they diverge by approaching 
$\lambda_{c}$
 as 
$V_{kl}∼{\left(\lambda_{c}-\lambda \right)}^{-1}$
. Figure 3b shows the exact result for the experimentally observable mean boson numbers 
$\langle {\hat{n}}_{x}\rangle$
 and 
$\langle {\hat{n}}_{y}\rangle$
. In the steady state regime, these two quantities are given by (see the Appendix A for details)

(11)
$$\begin{array}{ccc} & & {\langle {\hat{n}}_{x}\rangle }_{\mathrm{ss}}=\frac{\lambda^{4}\lambda_{c}^{2}\left(\lambda_{c}^{4}-\lambda^{4}\right)+2{\tilde{f}}^{2}\lambda_{c}^{6}}{8{\left(\lambda_{c}^{4}-\lambda^{4}\right)}^{2}},\\ & & {\langle {\hat{n}}_{y}\rangle }_{\mathrm{ss}}=\frac{\lambda^{4}\lambda_{c}^{2}\left(\lambda_{c}^{4}-\lambda^{4}+2{\tilde{f}}^{2}\right)}{8{\left(\lambda_{c}^{4}-\lambda^{4}\right)}^{2}}, \end{array}$$

where 
$\tilde{f}=f/\omega$
. As we see, the analytical results (Equation 11) match the exact result very closely. For 
$\tilde{f}\neq 0$
, the two quantities diverge as 
$\langle {\hat{n}}_{\beta }\rangle ∼{\left(\lambda_{c}-\lambda \right)}^{2}$
.

Let us now discuss the effect of the spin dephasing on the steady state. Such a decoherence effect can be described by including a spin dephasing term in Equation (Equation 4) with the jump operator 
${\hat{L}}_{\mathrm{dec}}=\sqrt{\frac{\Gamma }{2}}\sigma_{z}$
. Here, 
$\Gamma =1/\tau_{\mathrm{dep}}$
 stands for the constant dephasing rate, and 
$\tau_{\mathrm{dep}}$
 is the decoherence time. Since the periodic drives creates a spin-dependent coupling between the bosonic modes, one can expect that the spin decoherence would decrease the estimation precision. Remarkably, because the high-frequency drive causes a spin-dependent bosonic interaction, and the dissipative dynamics are time-independent, the resulting effective Liouvillian is diagonal in the spin basis such that the JT system becomes immune against the spin dephasing. We emphasize that the spin noisy decoupling is intimately related with the strong periodic drive in the JT system. We observe that, even in the presence of spin dephasing, the steady state numerical result for the average bosonic excitation in Figure 3b closely follow the analytical expression (Equation 11). In Figure 4a, the time evolution of the position quadrature and the mean boson number is shown, including the spin decoherence term in Equation (Equation 4), for 
$\gamma_{\beta }=0$
. Usually, the effect of the spin dephasing is to compromise the signal contrast. As we see, the high-frequency drive protects the signal contrast against spin dephasing. In Figure 4b, the position quadrature is plotted as a function of the dephasing rate 
Γ
 for different frequencies 
Φ
 and constant coupling 
λ
. We observe that, by increasing 
Φ
 and keeping 
λ
 fixed, one can further suppress the effect of the spin dephasing. This result indicates the periodic modulating JT system can serve as a probe for enhanced parameter estimation with measurement time not limited by spin dephasing.

Finally, I discuss the estimation precision using our periodically driven dissipative JT system. Since the steady state is in two-mode Gaussian form, one can characterize the sensitivity in terms of quantum Fisher information. For concreteness, I estimate the sensitivity of the force estimation. Because, in that case, all covariance matrix elements are independent of the parameter we wish to estimate, the corresponding quantum Fisher information is given by 
$F_{Q}\left(f\right)={\left(\partial_{f}d\right)}^{T}V{\left(\rho_{\mathrm{ss}}\right)}^{-1}\left(\partial_{f}d\right)$
 [29,30]. The force precision is bounded by the quantum Cramér-Rao bound, 
$\delta f^{2}\geq F_{Q}{\left(f\right)}^{-1}$
. I find

(12)
$$F_{Q}\left(f\right)=\frac{16\lambda_{c}^{6}+4\lambda_{c}^{4}\lambda^{4}-2\lambda^{8}}{\left(\lambda^{4}+4\left(\lambda_{c}^{2}-\lambda^{2}\right))\left(\lambda^{4}+4\left(\lambda_{c}^{2}+\lambda^{2}\right)\right)(\lambda_{c}^{4}-\lambda^{4}\right)}.$$

Approaching 
$\lambda \to \lambda_{c}$
, we have 
$F_{Q}\left(f\right)∼\left(1/2\lambda_{c}^{3}\right){\left(\lambda_{c}-\lambda \right)}^{-1}$
 so that the uncertainty in the force estimation becomes 
$\delta \tilde{f}∼\sqrt{2}\lambda_{c}^{3/2}{\left(\lambda_{c}-\lambda \right)}^{1/2}$
.

## 5. Physical Implementation

Trapped ions are suitable quantum system to implement the periodic modulating dissipative JT model by controlling sideband coupling with laser radiations [31]. Indeed, the time-dependent spin-boson interaction can be created using laser radiation, while, in order to realize the dissipative term in the Lindblad Equation (Equation 4), one needs to perform a sympathetic cooling of auxiliary ion. For this goal, I assume that two ions are confined in a linear Paul trap along the *z* axis with trap frequencies 
${\tilde{\omega }}_{x,y,z}$
, where the radial trap frequencies are much larger than the axial trap frequency 
$\omega_{x,y}\gg \omega_{z}$
, so that the ions are arranged in a linear configuration. I assume that ion 1 is used to implement the JT interaction, while ion 2 is the auxiliary ion which is not necessarily the same atomic species. In the limit of strong radial confinement, one can treat the small radial oscillations of the ions around their equilibrium positions in terms of local phonons. Then, the Hamiltonian which describes the *x*-*y* phonons becomes [32,33]

(13)
$${\hat{H}}_{\mathrm{ph}}=\sum_{\beta =x,y}\left\{\sum_{k=1}^{2}{\tilde{\omega }}_{\beta }{\hat{a}}_{k,\beta }^{†}{\hat{a}}_{k,\beta }+\kappa_{\beta }\left({\hat{a}}_{1,\beta }^{†}{\hat{a}}_{2,\beta }+{\hat{a}}_{1,\beta }{\hat{a}}_{2,\beta }^{†}\right)\right\}.$$

Here, 
${\tilde{\omega }}_{\beta }$
 is the local phonon frequency along the 
β
 direction, and 
$\kappa_{\beta }$
 is the Coulomb mediated hopping between sites 1 and 2. We assume that ion 1 has two metastable states 
$|↑\rangle$
 and 
$|↓\rangle$
 with transition frequency 
$\omega_{0}$
 such that the interaction free Hamiltonian is 
${\hat{H}}_{\mathrm{free}}={\hat{H}}_{\mathrm{ph}}+\left(\omega_{0}/2\right)\sigma_{z}$
.

Let us now discuss the physical implementation of the periodic JT interaction. Consider that ion 1 is simultaneously addressed by bichromatic laser fields along two transverse *x*-*y* directions with laser frequencies beat notes 
$\omega_{r,\beta }=\omega_{0}-\Delta -\left({\tilde{\omega }}_{\beta }-\omega_{\beta }\right)$
 and 
$\omega_{b,\beta }=\omega_{0}-\Delta +\left({\tilde{\omega }}_{\beta }-\omega_{\beta }\right)$
 which induce a transition between spin states 
$|↑\rangle$
 and 
$|↓\rangle$
. Here, 
Δ
 introduce effective spin frequency and 
$\omega_{\beta }$
 effective boson frequencies. The interaction Hamiltonian becomes

(14)
$$\begin{array}{ccc} {\hat{H}}_{I} & = & \Omega_{x}\left(t\right)\left\{\sigma^{+}e^{i\eta_{x}\left({\hat{a}}_{x}^{†}+{\hat{a}}_{x}\right)-i\phi_{x}}\left(e^{-i\omega_{r,x}t}+e^{-i\omega_{b,x}t}\right)+h.c.\right\}\\ & & +\Omega_{y}\left(t\right)\left\{\sigma^{+}e^{i\eta_{y}\left({\hat{a}}_{y}^{†}+{\hat{a}}_{y}\right)-i\phi_{y}}\left(e^{-i\omega_{r,y}t}+e^{-i\omega_{b,y}t}\right)+h.c.\right\}. \end{array}$$

Here, 
$\Omega_{x}\left(t\right)=2\Omega_{x,0}\cos\left(\Phi t\right)$
 and 
$\Omega_{x}\left(t\right)=2\Omega_{y,0}\sin\left(\Phi t\right)$
 are the time-dependent Rabi frequencies with amplitudes 
$\Omega_{\beta ,0}$
, 
$\phi_{\beta }$
 are the laser phases, and 
$\eta_{\beta }$
 are the Lamb-Dicke parameters. For simplicity, we denote 
${\hat{a}}_{1,\beta }={\hat{a}}_{\beta }$
. Next, we assume the Lamb-Dicke limit 
η≪1
 and transform the Hamiltonian (Equation 14) in the rotating-frame with respect to 
${\hat{U}}_{R}\left(t\right)=e^{-i\left(\omega_{0}-\Delta \right)t\left(\sigma_{z}/2\right)-i\sum_{\beta }\left\{\left({\tilde{\omega }}_{\beta }-\omega_{\beta }\right)t{\hat{a}}_{\beta }^{†}{\hat{a}}_{\beta }-i{\tilde{\omega }}_{\beta }t{\hat{a}}_{2,\beta }^{†}{\hat{a}}_{2,\beta }\right\}}$
, which yields

(15)
$${\hat{H}}_{0}+{\hat{H}}_{d}\left(t\right)={\hat{U}}_{R}^{†}\left({\hat{H}}_{\mathrm{free}}+{\hat{H}}_{I}\right){\hat{U}}_{R}-i{\hat{U}}_{R}^{†}\partial_{t}{\hat{U}}_{R},$$

where the spin-phonon couplings are 
$g_{\beta }=\eta_{\beta }\Omega_{\beta ,0}$
, and we assume that 
$\phi_{x}=\pi /2$
 and 
$\phi_{y}=0$
.

The dissipative dynamics of ion 1 can be implemented by performing a sympathetic cooling of the auxiliary ion 2 [34,35]. We assume that the auxiliary ion is continuously laser cooled with 
$\sum_{\beta }\hat{D}\left[{\hat{L}}_{\beta }\right]\hat{\rho }\left(t\right)$
, where the jump operators are 
${\hat{L}}_{\beta }=\sqrt{\Gamma_{\beta }}{\hat{a}}_{2,\beta }$
, with 
$\Gamma_{\beta }$
 being the cooling rates. The Heisenberg equation for the auxiliary *x*-*y* phonons becomes 
$\partial_{t}{\hat{a}}_{2,\beta }=i\kappa_{\beta }{\hat{a}}_{\beta }-\Gamma_{\beta }{\hat{a}}_{2,\beta }$
. In the limit 
$\Gamma_{\beta }\gg \kappa_{\beta }$
, one can adiabatically eliminate auxiliary modes, which gives an effective dissipative dynamics for ion 1 with rates 
$\gamma_{\beta }=\kappa_{\beta }^{2}/\Gamma_{\beta }$
.

## 6. Conclusions

I have proposed a quantum metrology application of the finite size periodic modulating JT model which describes the interaction between a single spin and two bosonic modes. The periodic modulating spin-boson couplings drive the system into a regime dictated by the time-independent effective Hamiltonian. In the high-frequency drive regime, the effective Hamiltonian describes a spin-dependent interaction between the two bosonic modes. I have shown that the energy gap vanishes at the critical point which can be used to enhance the precision of the parameter estimation. In particular, I have shown that the arbitrarily large boson or spin frequency estimation precision can be achieved close to a critical point.

Furthermore, I have discussed the effect of the loss of bosonic excitations on the time-dependent JT dynamics. The interplay between the periodic modulation and the dissipation drives the system into a nonequilibrium steady state. In the high-frequency drive regime, the time-evolution of the dissipative JT system is described in terms of an effective Liouvillian. I have shown that the steady state density matrix reviews a non-analytical behavior at the critical point, which can be used for high-precision parameter estimation. The key advantage of using periodic modulating JT quantum probe is the robustness against the spin dephasing. I have shown that, due to the high-frequency drive, the effective Liouvillian is diagonal in the spin basis, which makes the JT system immune against spin decoherence. Thanks to this, our frequency measurement time is not limited by the spin decoherence.

I have discussed the physical implementation of our model using trapped ions. The JT spin-boson couplings are created by applying bichromatic laser fields along the transverse directions with time-periodic intensity which couple the internal ion’s spin states and phonons. The driven-dissipative dynamics can be implemented by using an auxiliary ion which is continuously laser cooled. Finally, I note that our periodic modulating sensing technique is also relevant for other experimental setups, such as cavity or circuit QED systems [18,36].

## Figures and Tables

**Figure 1 entropy-23-01333-f001:**
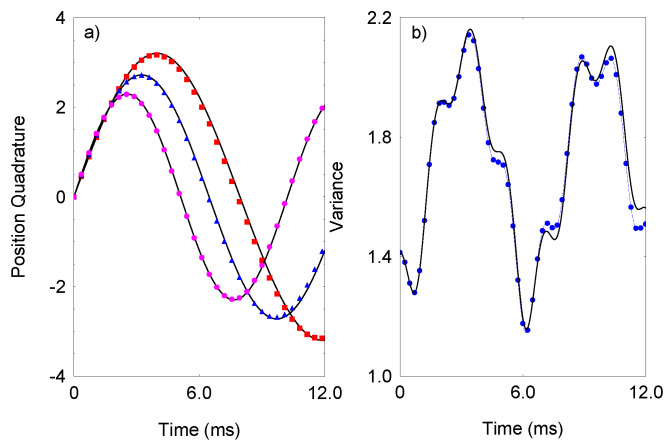
(Color online) (**a**) Time-evolution of the position quadrature 
$\langle \hat{x}\left(t\right)\rangle$
. We compare the exact result of the time-dependent Schrödinger equation with Hamiltonian (Equation 2) with the analytical expression (Equation 8) (solid lines) for 
λ=0.9
 (purple circles), 
λ=0.93
 (blue triangles), and 
λ=0.95
 (red squares). The other parameters are set to 
Δ=0
, 
g/2π=5.0
 kHz, and 
Φ/2π=1.1
 MHz. (**b**) Variance 
Δx(t)
 of the position quadrature. The exact result (blue circles) is compared with the analytical expression (Equation 8) (solid line) for 
λ=0.93
.

**Figure 2 entropy-23-01333-f002:**
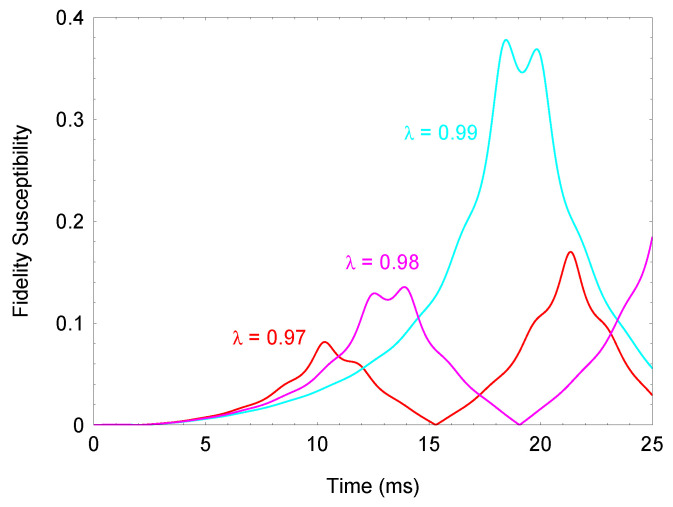
(Color online) Fidelity susceptibility 
$F_{x}=\partial_{\omega }\langle \hat{x}\left(t\right)\rangle /\Delta \hat{x}\left(t\right)$
 as a function of time for different couplings 
λ
. The parameters are 
g/2π=5.0
 kHz and 
Φ/2π=1.1
 MHz. Higher frequency sensitivity is achieved by increasing 
λ
 toward the critical value.

**Figure 3 entropy-23-01333-f003:**
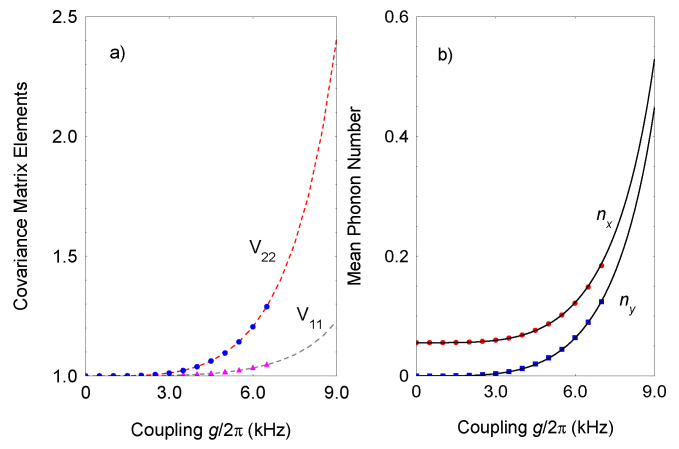
(Color online) (**a**) Covariance matrix elements 
$V_{11}$
 and 
V22
 as a function of the coupling strength 
$g=g_{x}=g_{y}$
. I compare the numerical solution of the time-dependent Liouvillian equation with Hamiltonian (Equation 2) blue circles and purple triangles with the solution using the time-average Liouvillian (Equation 6) with effective Hamiltonian (Equation 7) (dashed lines). The parameters are set to 
ω/2π=0.2
 kHz, 
Δ/2π=0.5
 kHz, 
Φ/2π=800
 kHz and 
γ/2π=0.5
 kHz. (**b**) Exact numerical result for the mean excitations of the two bosonic modes compared with the steady state analytical result (Equation 11). We set 
$\tilde{f}=1.27$
 and dephasing rate 
Γ/2π=2.5
 kHz.

**Figure 4 entropy-23-01333-f004:**
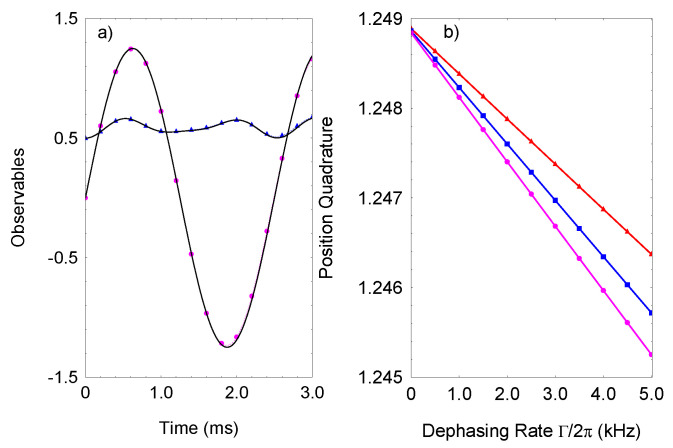
(Color online) (**a**) Position quadrature and mean boson as a function of time in the presence of spin dephasing. I compare the exact solution of the Lindblad equation with Hamiltonian (Equation 2) for 
$\langle \hat{x}\left(t\right)\rangle$
 (pink circles) and 
$\langle {\hat{n}}_{x}\left(t\right)\rangle$
 (blue triangles) with those given by the coherent evolution without spin dephasing (solid lines). The parameters are set to 
ω/2π=0.5
 kHz, 
Φ/2π=1.4
 MHz, 
Δ=0
, 
γ=0
, 
Γ/2π=2.0
 kHz, and 
λ=0.6
. (**b**) Exact result for the position quadrature as a function of the spin dephasing rate 
Γ
 at time 
$t=\pi /(2\omega \nu_{1})$
. We assume 
Φ/2π=1.4
 MHz (pink circles), 
Φ/2π=1.6
 MHz (blue circles), 
Φ/2π=2.0
 MHz (red triangles), and 
λ=0.6
.

## Data Availability

Not applicable.

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
