# Peer review of "Enhanced Parameter Estimation with Periodically Driven Quantum Probe"

_entropy, 2021, doi:10.3390/e23101333_

Round 1
Reviewer 1 Report
In this work, P Ivanov studied quadrature and frequency measurement using a spin-phonon model. This is a Jahn-Teller model of a single spin coupled to two phonon modes. The author claims that this model exhibits a "phase transition" by varying parameters in the model. It was found that the sensitivity of the measured quantities can be enhanced near the phase transition. In my opinion, this work presents interesting results, supported by numerical and analytical calculations. The analysis is thorough and careful. The work is relevant to the current study on using critical dynamics to estimate weak quantities, which attracts a growing attention in the physics community. The work should be published after considering the comments.
1. The phonon frequency is extremely low, which might be a problem in ion experiments. At this low frequency, the phonon temperature might be high and the typical Lamb-Dicke regime is difficult to achieve. Note laser cooling is not efficient or even possible due to the low phonon frequency.
2. I want to remind the author it was shown recently that a JT like model can be realized with trapped Rydberg ions, see Phys. Rev. Lett. 126, 233404, 2021.
Author Response
Point 1. The phonon frequency is extremely low, which might be a problem in ion experiments. At this low frequency, the phonon temperature might be high and the typical Lamb-Dicke regime is difficult to achieve. Note laser cooling is not efficient or even possible due to the low phonon frequency.
Response 1. I am agree with the referee that approaching the critical coupling the mean phonon excitation increases which can break the Lamb-Dicke approximation. One way to solve the problem is to use the magnetic field gradient which couples the spin and the two bosonic modes. In that case the high-order terms in the Lamb-Dicke approximation are zero and our sensing technique can be applied.
Point 2. I want to remind the author it was shown recently that a JT like model can be realized with trapped Rydberg ions, see Phys. Rev. Lett. 126, 233404, 2021.
Response 2. I thank the referee for point out this work. In the revised version of the manuscript I have included.
Reviewer 2 Report
The paper addresses in an original way the important problem of decoherence and dephasing in systems useful for quantum computing.
The theoretical approach is valid and contributes ti the search of a solution of this fundamental problem
Author Response
Point 1. The paper addresses in an original way the important problem of decoherence and dephasing in systems useful for quantum computing.
The theoretical approach is valid and contributes to the search of a solution of this fundamental problem .
Response 1. I thank the referee for the positive evaluation of the manuscript.
Reviewer 3 Report
The authors have described a new method for Enhanced Parameter Estimation with Periodically Driven Quantum Probe. The article has been divided into three main ideas in general. First, the article explains the interaction between a single spin and two 244 bosonic modes that have been described very well. Second, the authors explain the effect of the loss of bosonic excitations on the 252 time-dependent JT dynamics. Finally, the authors explain the physical implementation of the modes onto the JT dynamics.
As a result, the authors have represented a very good paper. I have a minor suggestion for the authors as following
1- the English language needs to be revised
2- references also need to be revised and add more
3- revise the abstract
Author Response
Point 1.
As a result, the authors have represented a very good paper. I have a minor suggestion for the authors as following
1- the English language needs to be revised.
2- references also need to be revised and add more.
3- revise the abstract.
Response 1. I thank the referee for the positive evaluation of the manuscript. I have included new references and also the text is improved.